# Cluster Variational Approximations for Structure Learning of Continuous-Time Bayesian Networks from Incomplete Data

**Dominik Linzner**[1] **and Heinz Koeppl**[1,2]
[1]Department of Electrical Engineering and Information Technology
[2]Department of Biology
Technische Universität Darmstadt
`{dominik.linzner, heinz.koeppl}@bcs.tu-darmstadt.de`

## Abstract

Continuous-time Bayesian networks (CTBNs) constitute a general and powerful framework for modeling continuous-time stochastic processes on networks. This makes them particularly attractive for learning the directed structures among interacting entities. However, if the available data is incomplete, one needs to simulate the prohibitively complex CTBN dynamics. Existing approximation techniques, such as sampling and low-order variational methods, either scale unfavorably in system size, or are unsatisfactory in terms of accuracy. Inspired by recent advances in statistical physics, we present a new approximation scheme based on cluster variational methods that significantly improves upon existing variational approximations. We can analytically marginalize the parameters of the approximate CTBN, as these are of secondary importance for structure learning. This recovers a scalable scheme for direct structure learning from incomplete and noisy time-series data. Our approach outperforms existing methods in terms of scalability.

## 1 Introduction

Learning directed structures among multiple entities from data is an important problem with broad applicability, especially in biological sciences, such as genomics [1] or neuroscience [20]. With prevalent methods of high-throughput biology, thousands of molecular components can be monitored simultaneously in abundance and time. Changes of biological processes can be modeled as transitions of a latent state, such as expression or non-expression of a gene, or activation/inactivation of protein activity. However, processes at the bio-molecular level evolve across vastly different time-scales [12]. Hence, tracking every transition between states is unrealistic. Additionally, biological systems are, in general, strongly corrupted by measurement- or intrinsic noise.

In previous numerical studies, continuous-time Bayesian networks (CTBNs) [13] have been shown to outperform competing methods for reconstruction of directed networks, such as ones based on Granger causality or the closely related dynamic Bayesian networks [1]. Yet, CTBNs suffer from the curse of dimensionality, prevalent in multi-component systems. This becomes problematic if observations are incomplete, as then the latent state of a CTBN has to be laboriously estimated [15]. In order to tackle this problem, approximation methods through sampling [8, 7, 19], or variational approaches [5, 6] have been investigated. These, however, either fail to treat high-dimensional spaces because of sample sparsity, are unsatisfactory in terms of accuracy, or provide good accuracy at the cost of an only locally consistent description.

In this manuscript, we present, to the best of our knowledge, the first direct structure learning method for CTBNs based on variational inference. We extend the framework of variational inference for

multi-component Markov chains by borrowing results from statistical physics on cluster variational methods [23, 22, 17]. Here the previous result in [5] is recovered as a special case. We derive approximate dynamics of CTBNs in form of a new set of ordinary differential equations (ODEs). We show that these are more accurate than existing approximations. We derive a parameter-free formulation of these equations, that depends only on the observations, prior assumptions, and the graph structure. Lastly, we recover an approximation for the structure score, which we use to implement a scalable structure learning algorithm. The notion of using marginal CTBN dynamics for network reconstruction from noisy and incomplete observations was recently explored in [21] to successfully reconstruct networks of up to eleven nodes by sampling from the exact marginal posterior of the process, albeit using large computational effort. Yet, the method is sampling-based and thus still scales unfavorably in high dimensions. In contrast, we can recover the marginal CTBN dynamics at once, using a standard ODE solver.

## 2 Background

### 2.1 Continuous-time Bayesian networks

We consider continuous-time Markov chains (CTMCs) $\{X(t)\}_{t \geq 0}$ taking values in a countable state-space $\mathcal{S}$. A time-homogeneous Markov chain evolves according to an intensity matrix $R : \mathcal{S} \times \mathcal{S} \to \mathbb{R}$, whose elements are denoted by $R(s, s')$, where $s, s' \in \mathcal{S}$. A continuous-time Bayesian network [13] is defined as an $N$-component process over a factorized state-space $\mathcal{S} = \mathcal{X}_1 \times \cdots \times \mathcal{X}_N$ evolving jointly as a CTMC. For local states $x_n, x'_n \in \mathcal{X}_n$, we will drop the states' component index $n$, if evident by the context and no ambiguity arises. We impose a directed graph structure $\mathcal{G} = (V, E)$, encoding the relationship among the components $V \equiv \{V_1, \ldots, V_N\}$, which we refer to as nodes. These are connected via an edge set $E \subseteq V \times V$. This quantity – the structure – is what we will later learn. The instantaneous state of each component is denoted by $X_n(t)$ assuming values in $\mathcal{X}_n$, which depends only on the states of a subset of nodes, called the parent set $\mathrm{pa}(n) \equiv \{m \mid (m, n) \in E\}$. Conversely, we define the child set $\mathrm{ch}(n) \equiv \{m \mid (n, m) \in E\}$. The dynamics of a local state $X_n(t)$ are described as a Markov process conditioned on the current state of all its parents $U_n(t)$ taking values in $\mathcal{U}_n \equiv \{\mathcal{X}_m \mid m \in \mathrm{pa}(n)\}$. They can then be expressed by means of the conditional intensity matrices (CIMs) $R_n^u : \mathcal{X}_n \times \mathcal{X}_n \to \mathbb{R}$, where $u_n \equiv (u_1, \ldots u_L) \in \mathcal{U}_n$ denotes the current state of the parents ($L = |\mathrm{pa}(n)|$). Specifically, we can express the probability of finding node $n$ in state $x'$ after some small time-step $h$, given that it was in state $x$ at time $t$ with $x, x' \in \mathcal{X}_n$ as

$$P(X_n(t + h) = x' \mid X_n(t) = x, U_n(t) = u) = \delta_{x,x'} + R_n^u(x, x')h + o(h),$$

where $R_n^u(x, x')$ is the matrix element of $R_n^u$ corresponding to the transition $x \to x'$ given the parents' state $u \in \mathcal{U}_n$. It holds that $R_n^u(x, x) = -\sum_{x' \neq x} R_n^u(x, x')$. The CIMs are connected to the joint intensity matrix $R$ of the CTMC via amalgamation – see, for example, [13].

### 2.2 Variational lower bound

The foundation of this work is to derive a variational lower bound on the evidence of the data for a CTMC. Such variational lower bounds are of great practical significance and pave the way to a multitude of approximate inference methods (*variational inference*). We consider paths $X_{[0,T]} \equiv \{X(\xi) \mid 0 \leq \xi \leq T\}$ of a CTMC with a series of noisy state observations $Y \equiv (Y^0, \ldots, Y^I)$ at times $(t^0, \ldots, t^I)$, drawn according to an observation model $Y^i \sim P(Y^i \mid X(t^i))$. We consider the posterior Kullback–Leibler (KL) divergence $D_{KL}(Q(X_{[0,T]}) || P(X_{[0,T]} \mid Y))$ given a candidate distribution $Q(X_{[0,T]})$, which can be decomposed as $D_{KL}(Q(X_{[0,T]}) || P(X_{[0,T]} \mid Y)) = D_{KL}(Q(X_{[0,T]}) || P(X_{[0,T]})) - \mathbb{E}[\ln P(Y \mid X_{[0,T]})] + \ln P(Y)$, where $\mathbb{E}[\cdot]$ denotes the expectation with respect to $Q(X_{[0,T]})$, unless specified otherwise. As $D_{KL}(Q(X_{[0,T]}) || P(X_{[0,T]} \mid Y)) \geq 0$ this recovers a lower bound on the evidence

$$\ln P(Y) \geq \mathcal{F}, \tag{1}$$

where the variational lower bound $\mathcal{F} \equiv -D_{KL}(Q(X_{[0,T]}) || P(X_{[0,T]})) + \mathbb{E}[\ln P(Y \mid X_{[0,T]})]$ is also known as the Kikuchi functional [11]. The Kikuchi functional has recently found heavy use in variational approximations for probabilistic models [23, 22, 17], because of the freedom it provides for choosing clusters in space and time. We will now make use of this feature.

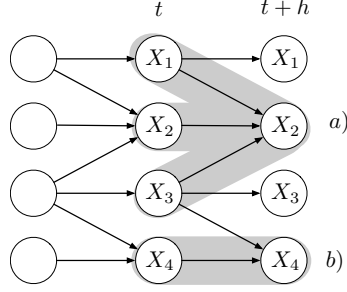

Figure 1: Sketch of different cluster choices for a CTBN in discretized time: a) star approximation b) naive mean-field.

## 3 Cluster variational approximations for CTBNs

The idea behind cluster variational approximations, derived subsequently, is to find an approximation of the variational lower bound using $M$ cluster functionals $\mathcal{F}_j$ of smaller sub-graphs $A_j(t)$ for a CTBN using its $h$-discretization (see Figure 1)

$$\mathcal{F} \approx \int_0^T \mathrm{d}t \sum_{j=1}^M \mathcal{F}_j(A_j(t)).$$

Examples for $A_j(t)$ are the completely local *naive mean-field approximation* $A_j^{\mathrm{mf}}(t) = \{X_j(t+h), X_j(t)\}$, or the *star approximation* $A_j^{\mathrm{s}}(t) = \{X_j(t+h), U_j(t), X_j(t)\}$ on which our method is based. In order to lighten the notation, we define $Q^h(s' \mid s) \equiv Q(X(t+h) = s' \mid X(t) = s)$ and $Q(s, t) \equiv Q(X(t) = s)$ for $Q$ and $P$, respectively. Marginal probabilities of individual nodes carry their node index as a subindex. The formulation of CTBNs imposes structure on the transition matrix

$$P^h(s' \mid s) = \prod_{n=1}^N P_n^h(x_n' \mid x_n, u_n), \qquad (2)$$

suggesting a node-wise factorization to be a natural choice. In order to arrive at the variational lower bound in the star approximation, we assume that $Q(X_{[0,T]})$ describes a CTBN, i.e. its transition matrices satisfy (2). However, to render our approximation tractable, we further restrict the set of approximating processes. Specifically, we require the existence of some expansion in orders of the *coupling strength* $\varepsilon$,

$$Q_n^h(x' \mid x, u) = Q_n^h(x' \mid x) + \mathcal{O}(\varepsilon) \quad \forall n \in \{1, \dots, N\}, \qquad (3)$$

where the remainder $\mathcal{O}(\varepsilon)$ contains the dependency on the parents.[1] In the following, we derive the star approximation of a factorized stochastic process. While the star approximation can be constructed according to the rules of cluster variational methods, see [23, 22, 17], we present a novel derivation via a perturbative expansion of the lower bound. This is meaningful as in cluster variational approximations, the assumptions on the approximating process (and similarity measures) and thus the resulting approximation error cannot be quantified analytically [23]. This new derivation also highlights the difference to conventional mean-field approximations, where only the class of approximating distributions is restricted. The exact expression of the variational lower bound $\mathcal{F}$ for a continuous-time Markov process decomposes into time-wise components $\mathcal{F} = \lim_{h \to 0} \frac{1}{h} \int_0^T \mathrm{d}t \, f^h(t)$

$$f^h(t) = \underbrace{\sum_{s',s} Q^h(s' \mid s)Q(s, t) \ln P^h(s' \mid s)}_{\equiv E(t)} - \underbrace{\sum_{s',s} Q^h(s' \mid s)Q(s, t) \ln Q^h(s' \mid s)}_{\equiv H(t)},$$

where we identified the time-dependent energy $E(t)$ and the entropy $H(t)$. Following (2), we can write $Q^h(s' \mid s) = \prod_n Q_n^h(x_n' \mid x_n, u_n)$. For now, we consider the time-dependent energy

$$E(t) = \sum_{s,s'} Q(s, t) \prod_n Q_n^h(x_n' \mid x_n, u_n) \ln \prod_k P_k^h(x_k' \mid x_k, u_k).$$

We start by making use of the assumption in (3). Subsequently, we arrive at an expansion of the energy by using the formula from Appendix B.1,

$$E(t) = \sum_{s,s'} Q(s,t) \prod_m Q_m^h(x_m' \mid x_m) \left\{ \sum_n \frac{Q_n^h(x_n' \mid x_n, u_n)}{Q_n^h(x_n' \mid x_n)} - (N-1) \right\}$$
$$\times \sum_k \ln P_k^h(x_k' \mid x_k, u_k) + \mathcal{O}(\varepsilon^2).$$

For each $k$, we can sum over $x'$ for each $n \neq k$. This leaves us with

$$E(t) = \sum_{s,s'} Q(s,t) \sum_n Q_n^h(x' \mid x, u) \ln P_n^h(x' \mid x, u) + \mathcal{O}(\varepsilon^2).$$

The exact same treatment can be done for the entropy term. Finally, assuming marginal independence $Q(s,t) = \prod_n Q_n(x_n,t)$, we arrive at the *weak coupling* expansion of the variational lower bound

$$f^h(t) = \sum_n \sum_{x',x \in \mathcal{X}_n} \sum_{u \in \mathcal{U}_n} Q_n^h(x' \mid x, u) Q_n(x,t) Q_n^u \ln \frac{P_n^h(x' \mid x, u)}{Q_n^h(x' \mid x, u)} + \mathcal{O}(\varepsilon^2),$$

with the shorthand $Q_n^u \equiv \prod_{l \in \mathrm{pa}(n)} Q_l(u_l, t)$. The variational lower bound $\mathcal{F}$ in star approximation (up to first order in $\varepsilon$) decomposes on the $h$-discretized network spanned by the CTBN process, into local star-shaped terms, see Figure 1. We emphasize that the variational lower bound in star approximation is no longer a lower bound on the evidence but provides an approximation. We note that in contrast to the naive mean-field approximation employed in [16, 5], we do not have to drop the dependence on the parents state of the variational transition matrix. Indeed, if we consider the variational lower bound in star approximation in zeroth order of $\varepsilon$, we recover exactly their previous result, demonstrating the generality of our method (see Appendix B.3).

## 3.1 CTBN dynamics in star approximation

We will now derive differential equations governing CTBN dynamics in star approximation. In order to perform the continuous-time limit $h \to 0$, we define,

$$\tau_n^u(x, x', t) \equiv \lim_{h \to 0} \frac{Q_n^t(x', x, u)}{h} \quad \text{for } x \neq x',$$

with the variational transition probability $Q_n^t(x', x, u) \equiv Q_n^h(x' \mid x, u) Q_n(x,t) Q_n^u$ and $\tau_n^u(x, x, t) \equiv -\sum_{x' \neq x} \tau_n^u(x, x', t)$. The variational transition probability can then be written as an expansion in $h$

$$Q_n^t(x', x, u) = \delta_{x,x'} Q_n(x,t) Q_n^u + h\tau_n^u(x, x', t) + o(h).$$

Checking self-consistency of this quantity via marginalization recovers an inhomogeneous Master equation

$$\dot{Q}_n(x,t) = \sum_{x' \neq x, u} [\tau_n^u(x', x, t) - \tau_n^u(x, x', t)]. \tag{4}$$

Because of the intrinsic asynchronous update constraint on CTBNs, only local probability flow inside the state-space $\mathcal{X}_n$ is allowed. This renders the above equation equivalent to a continuity constraint on the global probability distribution. After plugging in the variational transition probability into the variational lower bound, we arrive at a functional that is only dependent on the marginal distributions. Performing the limit of $h \to 0$, we recover at a sum of node-wise functionals in continuous-time (see Appendix B.2)

$$\mathcal{F} = \mathcal{F}_S + \mathcal{O}(\varepsilon^2), \quad \mathcal{F}_S \equiv \sum_{n=1}^N (H_n + E_n) + \mathcal{F}_0,$$

where we identified the variational lower bound in star approximation $\mathcal{F}_S$, the entropy $H_n$ and the energy $E_n$, respectively, as

$$H_n = \int_0^T dt \sum_{x,u} \sum_{x' \neq x} \tau_n^u(x, x', t) \left[ 1 - \ln \frac{\tau_n^u(x, x', t)}{Q_n(x,t) Q_n^u} \right],$$

$$E_n = \int_0^T dt \left[ \sum_x Q_n(x,t) \mathbb{E}_n[R_n^u(x,x)] + \sum_{x,u} \sum_{x' \neq x} \tau_n^u(x, x', t) \ln R_n^u(x, x') \right],$$

---

**Algorithm 1** Stationary points of Euler–Lagrange equation

---
1: **Input:** Initial trajectories $Q_n(x, t)$, boundary conditions $Q(x, 0)$ and $\rho(x, T)$ and data $Y$.
2: **repeat**
3:     **for all** $n \in \{1, \dots, N\}$ **do**
4:         **for all** $Y^i \in \mathbf{Y}$ **do**
5:             Update $\rho_n(x, t)$ by backward propagation from $t_i$ to $t_{i-1}$ using (5) fulfilling (6).
6:         **end for**
7:         Update $Q_n(x, t)$ by forward propagation using (4) given $\rho_n(x, t)$.
8:     **end for**
9: **until** Convergence
10: **Output:** Set of $Q_n(x, t)$ and $\rho_n(x, t)$.

---

and $\mathcal{F}_0 = \mathbb{E}[\ln P(Y \mid X_{[0,T]})]$. The *neighborhood average* is defined as $\mathbb{E}_n[f^u(x)] \equiv \sum_{u'} Q_n^{u'} f^{u'}(x)$ for any function $f^u(x)$. In principle, higher-order clusters can be considered [22, 17]. Lastly, we enforce continuity by (4) fulfilling the constraint. We can then derive the Euler–Lagrange equations corresponding to the Lagrangian,

$$\mathcal{L} = \mathcal{F}_S - \int_0^T \mathrm{d}t \sum_{n,x} \lambda_n(x, t) \left\{ \dot{Q}_n(x, t) - \sum_{x' \neq x, u} [\tau_n^u(x', x, t) - \tau_n^u(x, x', t)] \right\},$$

with Lagrange multipliers $\lambda_n(x, t)$.

The approximate dynamics of the CTBN can be recovered as stationary points of the Lagrangian, satisfying the Euler–Lagrange equation. Differentiating $\mathcal{L}$ with respect to $Q_n(x, t)$, its time-derivative $\dot{Q}_n(x, t)$, $\tau_n^u(x, x', t)$ and the Lagrange multiplier $\lambda_n(x, t)$ yield a closed set of coupled ODEs for the posterior process of the marginal distributions $Q_n(x, t)$ and transformed Lagrange multipliers $\rho_n(x, t) \equiv \exp(\lambda_n(x, t))$, eliminating $\tau_n^u(x, x', t)$,

$$\dot{\rho}_n(x, t) = \{\mathbb{E}_n[R_n^u(x, x)] + \psi_n(x, t)\}\rho_n(x, t) - \sum_{x' \neq x} \mathbb{E}_n[R_n^u(x, x')] \rho_n(x', t), \tag{5}$$

$$\dot{Q}_n(x, t) = \sum_{x' \neq x} Q_n(x', t)\mathbb{E}_n[R_n^u(x', x)] \frac{\rho_n(x, t)}{\rho_n(x', t)} - Q_n(x, t)\mathbb{E}_n[R_n^u(x, x')] \frac{\rho_n(x', t)}{\rho_n(x, t)}, \tag{6}$$

with

$$\psi_n(y, t) = \sum_{j \in \mathrm{ch}(n)} \sum_{x, x' \neq x} Q_j(x, t) \left\{ \frac{\rho_j(x', t)}{\rho_j(x, t)} \mathbb{E}_j[R_j^u(x, x') \mid y] + \mathbb{E}_j[R_j^u(x, x) \mid y] \right\},$$

where $\mathbb{E}_j[\cdot \mid y]$ for $y \in \mathcal{X}_n$ is the neighborhood average with the state of node $n$ being fixed to $y$. Furthermore, we recover the *reset condition*

$$\lim_{t \to t^{i-}} \rho_n(x, t) = \lim_{t \to t^{i+}} \rho_n(x, t) \exp \left\{ \sum_{s \in \mathcal{X} \mid s_n = x} \ln P(Y^i \mid s) \prod_{k=1, k \neq n}^N Q_k(x_k, t) \right\}, \tag{7}$$

which incorporates the conditioning of the dynamics on noisy observations. For the full derivation we refer the reader to Appendix B.4. We require boundary conditions for the evolution interval in order to determine a unique solution to the set of equations (5) and (6). We thus set either $Q_n(x, 0) = Y_n^0$ and $\rho_n(x, T) = Y_n^I$ in the case of noiseless observations, or – if the observations have been corrupted by noise – $Q_n(x, 0) = \frac{1}{2}$ and $\rho_n(x, T) = 1$ as boundaries before and after the first and the last observation, respectively. The coupled set of ODEs can then be solved iteratively as a fixed-point procedure in the same manner as in previous works [16, 5] (see Algorithm 1) in a forward-backward procedure. In order to incorporate noisy observations into the CTBN dynamics, we need to assume an observation model. In the following we assume that the data likelihood factorizes $P(Y^i \mid X) = \prod_n P(Y_n^i \mid X_n)$, allowing us to condition on the data by enforcing $\lim_{t \to t^{i-}} \rho_n(x, t) = \lim_{t \to t^{i+}} P_n(Y^i \mid x)\rho_n(x, t)$. In Figure 2, we exemplify CTBN dynamics ($N = 3$) conditioned on observations corrupted by independent Gaussian noise. We find close agreement with the exact posterior dynamics. Because we only need to solve $2N$ ODEs to approximate the dynamics of an $N$-component system, we recover a linear complexity in the number of components, rendering our method scalable.

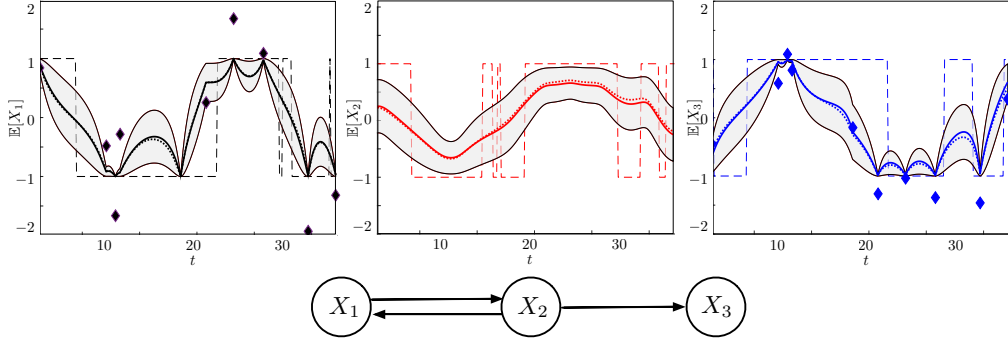

Figure 2: Dynamics in star approximation of a three node CTBN following Glauber dynamics at $a = 1$ and $b = 0.6$ conditioned on noisy observations (diamonds). We plotted the expected state (blue) plus variance (grey area). The observation model is the latent state plus gaussian random noise of variance $\sigma = 0.8$ and zero mean. The latent state (dashed) is well estimated for $X_2$, even when no data has been provided. For comparison, we plotted the exact posterior mean (dots). We did not plot the exact variance, which depends only on the mean, for better visibility.

## 3.2 Parameter estimation

Maximization of the variational lower bound with respect to transition rates $R_n^u(x, x')$ yields the expected result for the estimator of transition rates

$$\hat{R}_n^u(x, x') = \frac{\mathbb{E}[M_n^u(x, x')]}{\mathbb{E}[T_n^u(x)]},$$

given the *expected sufficient statistics* [15]

$$\mathbb{E}[T_n^u(x)] = \int_0^T \mathrm{d}t\, Q_n(x, t) Q_n^u, \quad \mathbb{E}[M_n^u(x, x')] = \int_0^T \mathrm{d}t\, \tau_n^u(x, x', t),$$

where $\mathbb{E}[T_n^u(x)]$ is the *expected dwelling time* for the $n$'th node in state $x$ and $\mathbb{E}[M_n^u(x, x')]$ are the *expected number of transitions* from state $x$ to $x'$, both conditioned on the parents state $u$. Following a standard *expectation–maximization* (EM) procedure, e.g. [16], we can estimate the systems' parameters given the underlying network.

## 3.3 Benchmark

In the following, we compare the accuracy of the star approximation with the naive mean-field approximation. Throughout this section, we will consider a binary local state-space (spins) $\mathcal{X}_n = \{+1, -1\}$. We consider a system obeying Glauber dynamics [10] with the rates $R_n^u(x, -x) = \frac{a}{2}\left(1 + x \tanh\left(b \sum_{l \in \mathrm{pa}(n)} u_l\right)\right)$. Here, $b$ is the inverse temperature of the system. With increasing $b$, the dynamics of each node depend more strongly on the dynamics of its neighbors. This corresponds to increasing the perturbation parameter $\varepsilon$. The pre-factor $a$ scales the overall rate of the process. This system is an appropriate toy-example for biological networks as it encodes additive threshold behavior. In Figure 3 $a)$ and $c)$, we show the mean-squared-error (MSE) between the expected sufficient statistics and the true ones for a tree network and an undirected chain with periodic boundaries of eight nodes, so that comparison with the exact result is still tractable. In this application, we restrict ourselves to noiseless observations to better connect to previous results as in [5]. We compare the estimation of the evidence using the variational lower bound in Figure 3 $b)$ and $d)$. We find that while our estimate using the star approximation is a much closer approximation, it does not provide a lower bound .

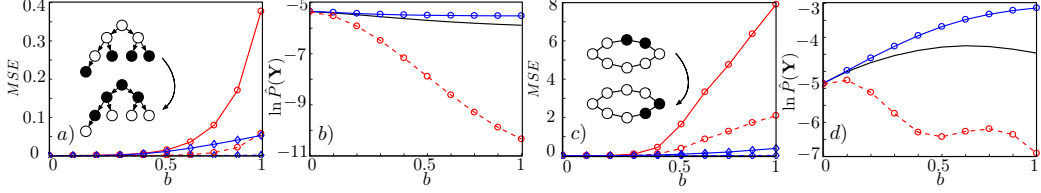

Figure 3: We perform inference on a tree network, see a) and b), and an undirected chain, displayed in c) and d). In both plots we consider CTBN of eight nodes with noiseless evidence as denoted in sketch inlet (black: $x = -1$, white: $x = 1$) in a) and c) obeying Glauber dynamics with $a = 8$. In a), we plot the mean-squared-error (MSE) for the expected dwelling times (dashed) and the expected number of transitions for the naive mean-field (circle, red) and star approximation (diamond, blue) with respect to the predictions of the exact simulation as a function of temperature $b$. In b) and d), we plot the approximation of logarithmic evidence as a function of temperature. We find that for both approximations (star approximation in blue, naive mean-field in red dashed and exact result in black) better performance on the tree network, while the star approximation clearly improves upon the naive mean-field approximation in both scenarios.

## 4 Cluster variational structure learning for CTBNs

For structure learning tasks, knowing the exact parameters of a model is in general unnecessary. For this reason, we will derive a parameter-free formulation of the variational approximation for the evidence lower bound and the latent state dynamics, analogous to the ones in the previous section. We derive an approximate CTBN structure score, for which we need to marginalize over the parameters of the variational lower bound. To this end, we assume that the parameters of the CTBN are random variables distributed according to a product of local and independent *Gamma distributions* $P(R \mid \boldsymbol{\alpha}, \boldsymbol{\beta}, \mathcal{G}) = \prod_n \prod_{x,u} \prod_{x' \neq x} Gam\left[R_n^u(x, x') \mid \alpha_n^u(x, x'), \beta_n^u(x)\right]$ given a graph structure $\mathcal{G}$. In star approximation, the evidence is approximately given by $P(Y \mid R, \mathcal{G}) \approx \exp(\mathcal{F}_S)$. By a simple analytical integration, we recover an approximation to the CTBN structure score

$$P(\mathcal{G} \mid Y, \boldsymbol{\alpha}, \boldsymbol{\beta}) \approx P(\mathcal{G}) \int_0^\infty \mathrm{d}R \, e^{\mathcal{F}_S} P(R \mid \boldsymbol{\alpha}, \boldsymbol{\beta}, \mathcal{G})$$

$$\propto e^H \prod_n \prod_{x,u} \prod_{x' \neq x} \left( \frac{\beta_n^u(x)}{(\mathbb{E}[T_n^u(x)] + \beta_n^u(x))^{M_n^u(x,x')}} \right)^{\alpha_n^u(x,x')} \frac{\Gamma\left(\mathbb{E}[M_n^u(x,x')] + \alpha_n^n(x,x')\right)}{\Gamma\left(\alpha_n^u(x,x')\right)}, \quad (8)$$

with $\Gamma$ being the Gamma-function. The approximated CTBN structure score still satisfies structural modularity, if not broken by the structure prior $P(\mathcal{G})$. However, an implementation of a *k-learn* structure learning strategy as originally proposed in [14] is prohibited, as the latent state estimation depends on the entire network. For a detailed derivation, see Appendix B.5. Finally, we note that, in contrast to the evidence in Figure 3, we have no analytical expression for the structure score (the integral is intractable), so that we can not compare with the exact result after integration.

### 4.1 Marginal dynamics of CTBNs

The evaluation of the approximate CTBN structure score requires the calculation of the latent state dynamics of the marginal CTBN. For this, we approximate the Gamma function in (8) via *Stirling's approximation*. As Stirling's approximation becomes accurate asymptotically, we imply that sufficiently many transitions have been recorded across samples or have been introduced via a sufficiently strong prior assumption. By extremization of the marginal variational lower bound, we recover a set of integro-differential equations describing the marginal self-exciting dynamics of the CTBN (see Appendix B.6). Surprisingly, the only difference of this parameter-free version compared to (5) and (6) is that the conditional intensity matrix has been replaced by its posterior estimate

$$\bar{R}_n^u(x, x') \equiv \frac{\mathbb{E}[M_n^u(x, x')] + \alpha_n^u(x, x')}{\mathbb{E}[T_n^u(x)] + \beta_n^u(x)}. \quad (9)$$

The rate $\bar{R}_n^u(x, x')$ is thus determined recursively by the dynamics generated by itself, conditioned on the observations and prior information. We notice the similarity of our result to the one recovered

Table 1: Experimental results with datasets generated from random CTBNs ($N = 5$) with families of up to $k_{max}$ parents. To demonstrate that our score prevents over-fitting we search for families of up to $k = 2$ parents. When changing one parameter the other default values are fixed to $D = 10$, $b = 0.6$ and $\sigma = 0.2$.

| $k_{max}$ | Experiment | Variable | AUROC | AUPR |
|---|---|---|---|---|
| 1 | Number of trajectories | $D = 5$ | $0.78 \pm 0.03$ | $0.64 \pm 0.01$ |
| | | $D = 10$ | $0.87 \pm 0.03$ | $0.76 \pm 0.00$ |
| | | $D = 20$ | $0.96 \pm 0.02$ | $0.92 \pm 0.00$ |
| | Measurement noise | $\sigma = 0.6$ | $0.81 \pm 0.10$ | $0.71 \pm 0.00$ |
| | | $\sigma = 1.0$ | $0.69 \pm 0.07$ | $0.49 \pm 0.01$ |
| 2 | Number of trajectories | $D = 5$ | $0.64 \pm 0.09$ | $0.50 \pm 0.17$ |
| | | $D = 10$ | $0.68 \pm 0.12$ | $0.54 \pm 0.14$ |
| | | $D = 20$ | $0.75 \pm 0.11$ | $0.68 \pm 0.16$ |
| | Measurement noise | $\sigma = 0.6$ | $0.71 \pm 0.13$ | $0.58 \pm 0.20$ |
| | | $\sigma = 1.0$ | $0.64 \pm 0.11$ | $0.53 \pm 0.15$ |

in [21], where, however, the expected sufficient statistics had to be computed self-consistently during each sample path. We employ a fixed-point iteration scheme to solve the integro-differential equation for the marginal dynamics in a manner similar to EM (for the detailed algorithm, see Appendix A.2).

## 5   Results and discussion

For the purpose of learning, we employ a *greedy hill-climbing* strategy. We exhaustively score all possible families for each node with up to $k$ parents and set the highest scoring family as the current one. We do this repeatedly until our network estimate converges, which usually takes only two of such *sweeps*. We can transform the scores to probabilities and generate *Reciever-Operator-Characteristics* (ROCs) and *Precision-Recall* (PR) curves by thresholding the averaged graphs. As a measure of performance, we calculate the averaged *Area-Under-Curve* (AUC) for both. We evaluate our method using both synthetic and real-world data from molecular biology [2]. In order to stabilize our method in the presence of sparse data, we augment our algorithm with a prior $\boldsymbol{\alpha} = 5$ and $\boldsymbol{\beta} = 10$, which is uninformative of the structure, for both experiments. We want to stress that, while we removed the bottleneck of exponential scaling of latent state estimation of CTBNs, Bayesian structure learning via scoring still scales super-exponentially in the number of components [9]. Our method can thus not be compared to shrinkage based network inference methods such as fused graphical lasso.

The synthetic experiments are performed on CTBNs encoded with Glauber dynamics. For each of the $D$ trajectories, we recorded 10 observations $Y^i$ at random time-points $t^i$ and corrupted them with Gaussian noise with variance $\sigma = 0.6$ and zero mean. In Table 1, we apply our method to random graphs consisting of $N = 5$ nodes and up to $k_{max}$ parents. We note that fixing $k_{max}$ does not fix the possible degree of the node (which can go up to $N - 1$). For random graphs with $k_{max} = 1$, our method performs best, as expected, and we are able to reliably recover the correct graph if enough data are provided. To demonstrate that our score penalizes over-fitting, we search for families of up to $k = 2$ parents. For the more challenging scenario of $k_{max} = 2$, we find a drop in performance. This can be explained by the presence of strong correlations in more connected graphs and the increased model dimension with larger $k_{max}$. In order to prove that our method outperforms existing methods in terms of scalability, we successfully learn a tree-network, with a leaf-to-root feedback, of 14 nodes with $a = 1$, $b = 0.6$, see Figure 4 II). This is the largest inferred CTBN from incomplete data reported (in [21] a CTBN of 11 nodes is learned, albeit with incomparably larger computational effort).

Finally, we apply our algorithm to the *In vivo Reverse-engineering and Modeling Assessment* (IRMA) network [4], a gene regulatory network that has been implemented on cultures of yeast, as a benchmark for network reconstruction algorithms, see Figure 4 I). Special care has been taken in order to isolate this network from crosstalk with other cellular components. It is thus, to best of our knowledge, the only molecular biological network with a ground truth. The authors of [4] provide time course data from two perturbation experiments, referred to as "switch on" and "switch off", and attempted

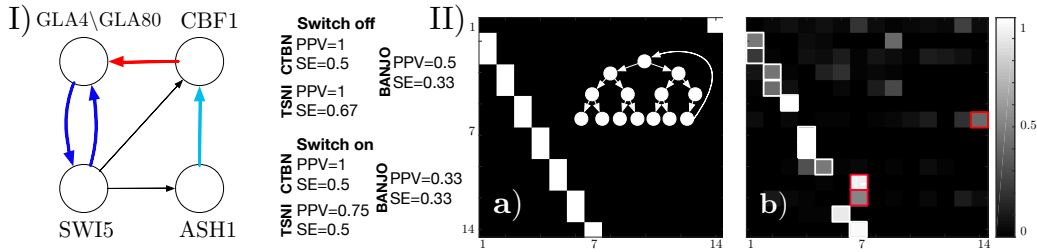

Figure 4: I) Reconstruction of a gene regulatory network (IRMA) from real-world data. To the left we show the inferred network for the "switch off" and "switch on" dataset. The ground truth network is displayed by black thin edges, the correctly inferred edges are thick (all inferred edges were correct). The the red edge was identified only in "switch on", the teal edge only in "switch off". On the right we show a small table summarizing the reconstruction capabilities of our method, TSNI and BANJO (PPV of random guess is 0.5). II) Reconstruction of large graphs. We tested our method on a ground truth graph with 14 nodes, as displayed in $a)$ with node-relations sketched in the inlet, encoded with Glauber dynamics and searched for a maximum of $k = 1$ parents. Although we used relatively few observations that have been strongly corrupted, the averaged learned graph $b)$ is visibly close to the ground truth. We framed the prediction of the highest scoring graph, where correctly learned edges are framed white and the incorrect ones are framed red.

reconstruction using different methods. In order to compare to their results we adopt their metrics *Positive Predicted Value* (PPV) and the *Sensitivity score* (SE) [2]. The best performing method is ODE-based (TSNI [3]) and required additional information on the perturbed genes in each experiment, which may not always be available. As can be seen in Figure 4 I), our method performs accurately on the "switch off" and the "switch on" data set regarding the PPV. The SE is slightly worse than for TSNI on "switch off". In both cases, we perform better than the other method based on Bayesian networks (BANJO [24]). Lastly, we note that in [1] more correct edges could be inferred using CTBNs, however with parameters tuned with respect to the ground truth to reproduce the IRMA network. For details on our processing of the IRMA data, see Appendix C. For comparison with other methods tested in [18] we refer to Appendix D where our method is consistently a top performer using AUROC and AUPR as metrics.

## 6  Conclusion

We develop a novel method for learning directed graphs from incomplete and noisy data based on a continuous-time Bayesian network. To this end, we approximate the exact but intractable latent process by a simpler one using cluster variational methods. We recover a closed set of ordinary differential equations that are simple to implement using standard solvers and retain a consistent and accurate approximation of the original process. Additionally, we provide a close approximation to the evidence in the form of a variational lower bound that can be used for learning tasks. Lastly, we demonstrate how marginal dynamics of continuous-time Bayesian networks, which only depend on data, prior assumptions, and the underlying graph structure, can be derived by the marginalization of the variational lower bound. Marginalization of the variational lower bound provides an approximate structure score. We use this to detect the best scoring graph using a greedy hill-climbing procedure. It would be beneficial to identify higher-order approximations of the variational lower bound in the future. We test our method on synthetic as well as real data and show that our method produces meaningful results while outperforming existing methods in terms of scalability.

## Acknowledgements

We thank the anonymous reviewers for helpful comments on the previous version of this manuscript. Dominik Linzner is funded by the European Union's Horizon 2020 research and innovation programme under grant agreement 668858.

## Footnotes

[1] An example of a function with such an expansion is a Markov random field with coupling strength $\varepsilon$.

[2]Our toolbox and code for experiments are available at `https://github.com/dlinzner-bcs/`.

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
