[Supplementary Material]

## Supplement: Cluster Variational Approximations for Structure Learning of Continuous-Time Bayesian Networks from Incomplete Data

In this supplementary, we give detailed descriptions on algorithms, data processing and derivations. Further, we provide an additional comparison to other methods for network reconstruction. All equation references point to the main text.

## A   Algorithms

---
**Algorithm 1** Marginal CTBN dynamics
---
1: **Input:** Propose set of initial trajectories $Q_n(x,t)$,
   observations $Y$, prior assumption on sufficient statistics $\boldsymbol{\alpha}$ and $\boldsymbol{\beta}$, initial guess for $\bar{R}_n^u(x,x')$.
2: **repeat**
3:    Set current $\bar{R}_n^u(x,x')$ as current CIM.
4:    Solve marginal dynamic equation with $\bar{R}_n^u(x,x')$ using Algorithm 1.
5:    Use expected sufficient statistics to update $\bar{R}_n^u(x,x')$ via (8).
6: **until** Convergence
7: **Output:** Set of $Q_n(x,t)$ and $\rho_n(x)$.

---

## B   Derivations

### B.1   Expansion formula

Note that for $\prod_{n=1}^N Q_n$ with $Q_n = a_n + \varepsilon b_n$ for any $a_n, b_n \in \mathbb{R}$ holds

$$\prod_{n=1}^N Q_n = \sum_{m=1}^N Q_m \prod_{n\neq m, n=1}^N a_n - (N-1)\prod_{n=1}^N a_n + \mathcal{O}(\varepsilon^2)$$

Proof:

$$\prod_{n=1}^N Q_n = \prod_{n=1}^N a_n + \epsilon \sum_{m=1}^N b_m \prod_{n\neq m, n=1}^N a_n + \sum_{m=1}^N a_m \prod_{n\neq m, n=1}^N a_n - \sum_{m=1}^N a_m \prod_{n\neq m, n=1}^N a_n + \mathcal{O}(\varepsilon^2)$$

$$= \sum_{m=1}^N [a_m + \epsilon b_m] \prod_{n\neq m, n=1}^N a_n + \prod_{n=1}^N a_n - \sum_{m=1}^N \prod_{n=1}^N a_n + \mathcal{O}(\varepsilon^2)$$

$$= \sum_{m=1}^N Q_m \prod_{n\neq m, n=1}^N a_n - (N-1)\prod_{n=1}^N a_n + \mathcal{O}(\varepsilon^2).$$

### B.2   Continuous-time variational lower bound in star approximation

In order to perform the continuous-time limit, we represent $Q$ as an expansion in $h$ in a set of marginals

$$Q_n^t(x', x, u) = \delta_{x,x'}Q_n(x,t)Q_n^u + h\tau_n^u(x,x',t) + o(h),$$

with $\tau_n^u(x,x,t) = -\sum_{x'\neq x} \tau_n^u(x,x',t)$. We note, that this corresponds to an additional mean-field assumption of marginal independence $Q(s,t) = \prod_n Q_n(x_n,t)$, however in contrast to the naive mean-field approximation in previous works [3, 2], we do not have to constrain $\tau_n^u(x,x',t)$ in order

to yield tractable results. By inserting $Q's$ representation into $\mathcal{F}$ we get

$$
\mathcal{F} = -\sum_n \lim_{h\to 0} \frac{1}{h} \int_0^T \mathrm{d}t \sum_{x,x'\neq x,u} h\tau_n^u(x,x',t) \left[ \ln h \frac{\tau_n^u(x,x',t)}{Q_n(x,t)Q_n^u} - \ln h R_n^u(x,x') \right]
$$

$$
- \sum_n \lim_{h\to 0} \frac{1}{h} \int_0^T \mathrm{d}t \sum_{x,u} \left[ Q_n^u Q_n(x,t) - h \sum_{x'\neq x} \tau_n^u(x,x',t) \right]
$$

$$
\times \left[ \ln \left( 1 - h \frac{\sum_{x'\neq x} \tau_n^u(x,x',t)}{Q_n(x,t)Q_n^u} \right) - \ln\left(1 + hR_n^u(x,x)\right) \right]
$$

where we also inserted $P_n^h(x' \mid x,u) = \delta_{x,x'} + hR_n^u(x,x')$. With the asymptotic identity $\ln(1+hx) = hx$ we can simplify

$$
\mathcal{F} = -\sum_n \lim_{h\to 0} \frac{1}{h} \int_0^T \mathrm{d}t \sum_{x,x'\neq x,u} h\tau_n^u(x,x',t) \left[ \ln \frac{\tau_n^u(x,x',t)}{Q_n(x,t)Q_n^u} - \ln R_n^u(x,x') \right]
$$

$$
+ \sum_n \lim_{h\to 0} \frac{1}{h} \int_0^T \mathrm{d}t \sum_{x,u} \left[ Q_n^u Q_n(x,t) - h \sum_{x'\neq x} \tau_n^u(x,x',t) \right] \left[ h \frac{\sum_{x'\neq x} \tau_n^u(x,x',t)}{Q_n(x,t)Q_n^u} + hR_n^u(x,x) \right]
$$

which becomes in the continuous-time limit $h \to 0$

$$
\mathcal{F} = \sum_n \int_0^T \mathrm{d}t \sum_{x,x'\neq x,u} \tau_n^u(x,x',t)[1 - \ln \tau_n^u(x,x',t) + \ln(Q_n^u Q_n(x,t))]
$$

$$
+ \sum_n \int_0^T \mathrm{d}t \left[ \sum_{x,u} Q_n(x,t)Q_n^u R_n^u(x,x) + \sum_{x,x'\neq x,u} \tau_n^u(x,x',t) \ln R_n^u(x,x') \right].
$$

The contribution of the likelihood term can be derived to be

$$
\mathcal{F}_0 = \sum_t h \sum_i \mathbb{E}[\ln P(Y^i \mid x)] \frac{\delta_{t,t^i}}{h} \underset{h\to 0}{=} \int_0^T \mathrm{d}t \sum_i \mathbb{E}[\ln P(Y^i \mid x)]\delta(t - t^i),
$$

$$
\mathbb{E}[f(x)] = \sum_{x\in\mathcal{X}} f(x) \prod_{k=1}^N Q_k(x_k), \quad x_k \in \mathcal{X}_k.
$$

## B.3   Relation to naive mean-field approximation

We recover the variational lower bound in naive mean-field approximation by only consider the zeroth order expansion in the correlations $\varepsilon$, meaning

$$
Q_n^h(x' \mid x,u) = Q_n^h(x' \mid x)
$$

Then for the energy $E(t)$ from the main text holds

$$
E(t) \equiv \sum_{s',s} Q(s,t)Q^h(s' \mid s) \ln P^h(s' \mid s)
$$

$$
= \sum_{s',s} \prod_n Q_n^t(x_n)Q_n^h(x_n' \mid x_n) \ln \left[ \prod_m P_m^h(x_m' \mid x_m, u_m) \right].
$$

Thus for the variational lower bound we arrive at the naive mean-field approximation

$$
\mathcal{F} = \sum_n \lim_{h\to 0} \frac{1}{h} \int_0^T \mathrm{d}t \sum_{x',x} Q_n^h(x' \mid x) \sum_u \prod_{k\in\mathrm{pa}(n)} Q_k(u_k,t) \ln \frac{P_n^h(x' \mid x,u)}{Q_n^h(x' \mid x)}
$$

Finally considering the marginals of the transitions

$$
Q(x' \mid x) = \delta_{x,x'} + h\frac{\tau_n(x,x',t)}{Q_n(x,t)} + o(h),
$$

and using an identical derivation as given in Appendix B.2

$$\mathcal{F} = \sum_n \lim_{h \to 0} \frac{1}{h} \int_0^T dt \sum_{x'_n, x_n} Q_n(x, t) \left[ \delta_{x_n, x'_n} + h \frac{\tau_n(x, x', t)}{Q_n(x, t)} \right]$$

$$\times \sum_u \prod_{k \in \mathrm{pa}(n)} Q_k(u_k, t) \ln \frac{\delta_{x_n, x'_n} + h R_n^u(x, x')}{\delta_{x_n, x'_n} + h \frac{\tau_n(x, x', t)}{Q_n(x, t)}}$$

$$= \sum_n \lim_{h \to 0} \frac{1}{h} \int_0^T dt \sum_{x_n, x'_n \neq x_n} h \tau_n(x, x', t) \sum_u \prod_{k \in \mathrm{pa}(n)} Q_k(u_k, t) \ln \frac{Q_n(x, t) R_n^u(x, x')}{\tau_n(x, x', t)}$$

$$+ \sum_n \lim_{h \to 0} \frac{1}{h} \int_0^T dt \sum_{x_n} Q_n(x, t) \left[ 1 + h \frac{\tau_n(x, x, t)}{Q_n(x, t)} \right]$$

$$\times \sum_u \prod_{k \in \mathrm{pa}(n)} Q_k(u_k, t) \ln \frac{1 + h R_n^u(x, x)}{1 + h \frac{\tau_n(x, x, t)}{Q_n(x, t)}}.$$

We can low evaluate $\lim_{h \to 0}$. Further we use the definition of the expectation from the main text and write

$$\mathcal{F} = \sum_n \int_0^T dt \sum_{x_n, x'_n \neq x_n} \tau_n(x, x', t) \left[ \mathbb{E}_n[\ln R_n^u(x, x')] - \ln \frac{\tau_n(x, x', t)}{Q_n(x, t)} \right]$$

$$+ \sum_n \int_0^T dt \sum_{x_n, x'_n \neq x_n} Q_n(x, t) h \left[ \mathbb{E}_n[R_n^u(x, x')] - \frac{\tau_n(x, x, t)}{Q_n(x, t)} \right].$$

After reordering terms and we recover the variational lower bound for CTBNs in naive mean-field approximation from literature [3, 2]

$$\mathcal{F} = \sum_n \int_0^T dt \sum_{x, x' \neq x} [Q_n(x, t) \mathbb{E}_n[R_n^u(x, x)] + \tau_n(x, x', t) \mathbb{E}_n[\ln R_n^u(x, x')]]$$

$$+ \sum_n \int_0^T dt \sum_{x, x' \neq x} \tau_n(x, x', t) \left[ 1 - \ln \frac{\tau_n(x, x', t)}{Q_n(x, t)} \right].$$

## B.4 CTBN dynamics in star approximation

We are now going to derive the dynamics of CTBNs in star approximation, defined by fulfilling the Euler–Lagrange equations

$$\partial_x \mathcal{L}[t, x, \dot{x}] - \partial_t [\partial_{\dot{x}} \mathcal{L}[t, x, \dot{x}]] = 0.$$

First lets consider the derivative with respect to $Q_n(x, t)$:

$$\partial_{Q_n(x, t)} H_n = \sum_u \sum_{x' \neq x} \frac{\tau_n^u(x, x', t)}{Q_n(x, t)}, \quad \partial_{Q_n(x, t)} E_j = \mathbb{E}_n[R_n^u(x, x)],$$

Further if node $n$ has a child $j$

$$\partial_{Q_n(x, t)} H_j = \sum_{x, u | X_n(t) = x_n = x} \sum_{x' \neq x} \frac{\tau_j^u(x, x')}{Q_n(x, t)}, \quad \partial_{Q_n(x, t)} E_j = \sum_x Q_j(x) \mathbb{E}_n[R_n^u(x, x) \mid X_n(t) = x].$$

With respect to the derivative $\dot{Q}_n(x, t)$ we get

$$\partial_{\dot{Q}_n(x, t)} \mathcal{L} = -\lambda_n(x, t).$$

We derive with respect to the transitions

$$\partial_{\tau_n^u(x, x', t)} H_n = \ln[Q_n(x, t) Q_n^u] - \ln \tau_n^u(x, x', t), \quad \partial_{\tau_n^u(x, x', t)} E_n = \ln R_n^u(x, x').$$

thus

$$\partial_{\tau_n^u(x,x',t)}\mathcal{L} = \ln[Q_n(x,t)Q_n^u] - \ln\tau_n^u(x,x',t) + \ln R_n^u(x,x') - \lambda_n(x,t) + \lambda_n(x',t).$$

The derivative with respect to the Lagrange-multipliers yields:

$$\partial_{\lambda_n(x,t)}\mathcal{L} = -\left\{\dot{Q}_n(x,t) - \left[\sum_{x'\neq x,u}\tau_n^u(x',x,t) - \tau_n^u(x,x',t)\right]\right\}$$

And lastly assuming a factorized noise model $P(Y^i|X(t) = s) = \prod_n P_n(Y^i|X_n(t) = x)$ we have for the derivative of $\mathcal{F}_0$

$$\partial_{Q_n(x,t)}\mathcal{F}_0 = \sum_i \ln P_n(Y^i|x)\delta(t - t^i)$$

These can then be combined as the following Euler-Lagrange equations:

(I) $\quad 0 = \sum_u \sum_{x'\neq x} \frac{\tau_n^u(x,x',t)}{Q_n(x,t)} + \mathbb{E}_n[R_n^u(x,x')] + \dot{\lambda}_n(x) + \sum_i \ln P_n(Y^i|x)\delta(t - t^i)$

$\qquad + \sum_{j\in\text{child}(n)}\sum_{x,u|X_n(t)=x}\sum_{x'\neq x} \frac{\tau_n^u(x,x',t)}{Q_n(x,t)} + \sum_x Q_j(x)\mathbb{E}_j[r_j^u(x,x) \mid X_n(t) = x]$

(II) $\quad 0 = \ln[Q_n(x,t)Q_n^u] - \ln\tau_n^u(x,x',t) + \ln R_n^u(x,x') - \lambda_n(x,t) + \lambda_n(x',t)$

(III) $\quad \dot{Q}_n(x,t) = \sum_{x'\neq x,u}\tau_n^u(x',x) - \tau_n^u(x,x',t).$

Exponentiating (II) gives

(II*) $\quad \tau_n^u(x,x',t) = Q_n(x,t)Q_n^u R_n^u(x,x')\rho_n(x',t)/\rho_n(x,t),$

where $\rho_n(x,t) \equiv \exp(\lambda_n(x,t))$. Assuming that $R$ is irreducible, $\rho_n(x,t)$ and $Q_n(x,t)$ are non-zero in $(0,T)$ and we can thus eliminate $\tau_n^u(x,x',t)$ in (I) and (II). Thus

(I*) $\quad \dot{\rho}_n(x,t) = \sum_{x'\neq x}\mathbb{E}_n[R_n^u(x,x')]\rho_n(x',t) + \{\mathbb{E}_n[R_n^u(x,x)] + \psi_n(x,t)\}\rho_n(x,t)$

(III*) $\quad \dot{Q}_n(x,t) = \sum_{x'\neq x}\{Q_n(x',t)\mathbb{E}_n[R_n^u(x',x)]\rho_n(x,t)/\rho_n(x',t)$

$\qquad\qquad\qquad -Q_n(x,t)\mathbb{E}_n[R_n^u(x,x')]\rho_n(x',t)/\rho_n(x,t)\},$

where we used $\dot{\lambda}_n(x,t) = \frac{1}{\rho_n(x,t)}\dot{\rho}_n(x,t)$. We further summarized

$$\psi_n(y,t) = \sum_{j\in\text{child}(n)}\sum_x Q_j(x)\left\{\sum_{x'\neq x}\frac{\rho_j(x',t)}{\rho_j(x,t)}\mathbb{E}_n[r_j^u(x,x') \mid y] + \mathbb{E}_n[r_j^u(x,x) \mid y]\right\}$$

$$+ \sum_i \ln P_n(Y^i|x)\delta(t - t^i).$$

The driving term $\ln P_n(Y^i|x)\delta(t - t^i)$ then conditions the dynamics on the observations by $\lim_{t\to t^{i-}}\rho_n(x,t) = \lim_{t\to t^{i+}}P_n(Y^i|x)\rho_n(x,t).$

## B.5 Variational marginal score

Using $R_n^u(x,x) = -\sum_{x'\neq x}R_n^u(x,x')$ we can write

$$E_n = \int dt \sum_{x,u}\sum_{x'\neq x}[\tau_n^u(x,x',t)\ln R_n^u(x,x') - Q_n(x,t)Q_n^u R_n^u(x,x')].$$

For the approximated evidence

$$P(Y \mid R,\mathcal{G}) \approx \prod_n \exp[E_n + H_n].$$

we get

$$P(Y \mid R, \mathcal{G}) \approx e^H \prod_n \prod_{x,u} \prod_{x' \neq x} R_n^u(x, x')^{\mathbb{E}[M_n^u(x,x')]} e^{-\mathbb{E}[T_n^u(x)]R_n^u(x,x')}.$$

Assuming an independent *Gamma* prior

$$P(R \mid \boldsymbol{\alpha}, \boldsymbol{\beta}, \mathcal{G}) = \prod_n \prod_{x,u} \prod_{x' \neq x} \gamma\left[r_n^u(x, x') \mid \alpha_n^u(x, x'), \beta_n^u(x)\right]$$

$$= \prod_n \prod_{x,u} \prod_{x' \neq x} \frac{\beta_n^u(x)^{\alpha_n^u(x,x')}}{\Gamma(\alpha_n^u(x,x'))} R_n^u(x, x')^{\alpha_n^u(x,x')-1} e^{-\beta_n^u(x)R_n^u(x,x')}.$$

Thus we can express the graph posterior

$$P(\mathcal{G}|Y, \boldsymbol{\alpha}, \boldsymbol{\beta}) \propto P(\mathcal{G}) \int_0^\infty P(Y \mid R, \mathcal{G}) P(R \mid \boldsymbol{\alpha}, \boldsymbol{\beta}, \mathcal{G}) \, \mathrm{d}R$$

$$\approx e^H \prod_n \prod_{x,u} \prod_{x' \neq x} \frac{\beta_n^u(x)^{\alpha_n^u(x,x')}}{\Gamma(\alpha_n^u(x,x'))}$$

$$\times \int_0^\infty R_n^u(x, x')^{\mathbb{E}[M_n^u(x,x')]+\alpha_n^u(x,x')-1} e^{-[\mathbb{E}[T_n^u(x)]+\beta_n^u(x)]R_n^u(x,x')} \mathrm{d}R_n^u(x, x'),$$

which has an analytic solution

$$\int_0^\infty x^a e^{-bx} \mathrm{d}x = b^{-a} \Gamma(a).$$

Thus we get

$$P(\mathcal{G} \mid Y, \boldsymbol{\alpha}, \boldsymbol{\beta}) \propto e^H \prod_n \prod_{x,u} \prod_{x' \neq x} \left( \frac{\beta_n^u(x)}{(\mathbb{E}[T_n^u(x)] + \beta_n^u(x))^{\mathbb{E}[M_n^u(x,x')]}} \right)^{\alpha_n^u(x,x')} \frac{\Gamma(\mathbb{E}[M_n^u(x,x')] + \alpha_n^u(x,x'))}{\Gamma(\alpha_n^u(x,x'))}.$$

### B.6   Marginal dynamics for CTBNs

In the following we are going to derive the dynamic equations of the marginal process for which we have to expand the Gamma-function. Assuming the sum of recorded transitions and prior transition number to be sufficiently large we can approximate the Gamma-function using Stirling's approximation $\Gamma(z) \approx \sqrt{\frac{2\pi}{z}} \left(\frac{z}{e}\right)^z + \mathcal{O}\left(\frac{1}{z}\right)$ we get the approximate marginal score function

$$\ln P(\mathcal{G} \mid Y, \boldsymbol{\alpha}, \boldsymbol{\beta}) \propto \sum_n H_n + \mathcal{E}_n,$$

with

$$\mathcal{E}_n = \sum_{x,u} \sum_{x' \neq x} \left( \mathbb{E}[M_n^u(x, x')] + \alpha_n^u(x, x') - \frac{1}{2} \right) \ln \left( \mathbb{E}[M_n^u(x, x')] + \alpha_n^u(x, x') \right)$$

$$- \left( \alpha_n^u(x, x') - \frac{1}{2} \right) \ln \left( \alpha_n^u(x, x') \right) + \alpha_n^u(x, x') \ln \left( \beta_n^u(x) \right)$$

$$- \left( \mathbb{E}[M_n^u(x, x')] + \alpha_n^u(x, x') \right) \ln \left( \mathbb{E}[T_n^u(x)] + \beta_n^u(x) \right) - \mathbb{E}[M_n^u(x, x')],$$

$$\partial_{Q_n(x,t)} \mathcal{E}_n = - \sum_{x' \neq x} \sum_u \mathbb{E}_n \left[ \frac{\mathbb{E}[M_n^u(x, x')] + \alpha_n^u(x, x')}{\mathbb{E}[T_n^u(x)] + \beta_n^u(x)} \right],$$

$$\partial_{Q_n(x,t)} \mathcal{E}_j = - \sum_x \sum_{x' \neq x} Q_j(x) \mathbb{E}_n \left[ \frac{\mathbb{E}[M_j^u(x, x')] + \alpha_j^u(x, x')}{\mathbb{E}[T_j^u(x)] + \beta_j^u(x)} \mid X_n(t) = x \right] \mathbb{1}[j \in \text{child}(n)],$$

$$\partial_{\tau_n^u(x,x',t)} \mathcal{E}_n \approx \ln \left( \frac{\mathbb{E}[M_n^u(x, x')] + \alpha_n^u(x, x')}{\mathbb{E}[T_n^u(x)] + \beta_n^u(x)} \right),$$

where we approximated $\frac{\mathbb{E}[M_n^u(x,x')]+\alpha_n^u(x,x')-\frac{1}{2}}{\mathbb{E}[M_n^u(x,x')]+\alpha_n^u(x,x')} - 1 \approx 0$. The derivatives with respect to $H_n$ and the constraint remain unchanged, see Appendix B.4. Finally defining the posterior-rate

$$\bar{R}_n^u(x,x') \equiv \frac{\mathbb{E}[M_n^u(x,x')] + \alpha_n^u(x,x')}{\mathbb{E}[T_n^u(x)] + \beta_n^u(x)},$$

we arrive at the same set of equations as in Appendix B.4.

## C   Processing IRMA data

In this section we present our approach of processing IRMA data. The IRMA dataset consists of expression data of genes, measured in concentrations, which are continuous. We can not capture continuous data using CTBNs, but need to map this data to a set of latent states. We identify two states *over-expressed* ($X = 1$) and *under-expressed* ($X = 0$) with respect to the *basal* (equilibrium) concentration $c_B$. This motivates the following observation model given the basal concentration

$$P(Y \mid X = 1, c_B) = \begin{cases} 1/|Y_0| & ,Y \geq c_B \text{ and } Y \leq Y_0 \\ 0 & ,\text{else} \end{cases},$$

$$P(Y \mid X = 0, c_B) = \begin{cases} 1/|Y_0| & ,Y < c_B \text{ and } Y \geq -Y_0 \\ 0 & ,\text{else} \end{cases},$$

where we have to choose some $Y_0$, so that the likelihood is normalized. We set $Y_0$ to some large value $Y_0 \geq \mathrm{argmax}_{|Y| \in \mathrm{DATA}}$ as our method remains invariant under each choice.

We model the basal concentration itself is a random variable, which we assume is gaussian distributed. We can estimate the parameters of the gaussian distribution $\mu_B$ and $\sigma_B$ from the data. The marginal observation model is then acquired by integration

$$P(Y \mid X) = \begin{cases} 1 - \mathrm{erf}((Y - \mu_B)/\sigma_B) & ,X = 1 \\ \mathrm{erf}((Y - \mu_B)/\sigma_B) & ,X = 0 \end{cases}.$$

Given this observation model we can assign each measurement a likelihood and can process the data using our method. We note that other models for IRMA data can be thought of that may return better (or worse) results using our method.

## D   Comparsion to other methods for network reconstruction

We compare our method to the methods for network reconstruction from time-series expression data considered in [4], see 1. These tests have, in contrast to [1], been performed on the full IRMA network. We adopt the shorthands of this paper to refer to different methods. The methods are based on dynamic Bayesian networks (DBNs), ODEs (TNSI), non-parametrics (NDS) and Granger Causality (GC). For more details on these methods we refer to [4]. Our method (CTBN CVM) outperforms all competing methods on the "switch on" dataset. On the "switch off" dataset the non-parametric CSI$^d$ method has a higher AUROC but a lower AUPR than our method.

Table 1: Comparison of AUROC (AUPR) of different methods on IRMA dataset. Results of top performers are in bold.

| method | | switch on | switch off |
|---|---|---|---|
| steady state | knockout | 0.68 (0.42) | 0.81 (0.50) |
| DBN | G1DBN | 0.78 (0.64) | 0.61 (0.34) |
| | VBSSM | 0.79 (0.70) | 0.76 (0.60) |
| ODE | TNSI | 0.68 (0.51) | 0.68 (0.42) |
| NDS | GP4GRN | 0.73 (0.61) | 0.76 (0.57) |
| | CSI$^d$ | 0.63 (0.46) | **0.86** (0.72) |
| | CSI$^c$ | 0.64 (0.39) | 0.73 (0.59) |
| GC | GCCA | 0.71 (0.55) | 0.74 (0.65) |
| CTBN | CVM | **0.82 (0.74)** | 0.84 (**0.76**) |
| random | | 0.65 (0.45) | 0.65 (0.45) |