[Reviews · NeurIPS 2018]

Reviewer 1



The paper introduces a generalization of previous variational methods for inference with jumps processes; here, the proposal approximating measure to the posterior relies on a star approximation. In application to continuous-time Bayesian networks, this means isolating clusters of nodes across children and parents, in order to build an efficient approximation to the traditional variational lower bound. The paper further presents examples and experiments that show how the proposed approach can be adapted to structure learning tasks in continuous-time settings. This is an interesting and topical contribution likely to appeal to the statistical and probabilistic community within NIPS. The paper is, in overall, well-written and reasonably well-structured. It offers a good background on previous work, helps the reader to understand its relevance and put its results in context within the existing literature. For these reasons, I wish to vote for weak acceptance. However, I have two major comments regarding rigor and content within some parts of the text. Specifically, 1 - The very first formulae presented around equation (1) seems wrong, unless I am missing something trivial. I pressume the negative lnP(Y) on the above equation has to be added, not substracted. As these are fairly common concepts and inequalities, this mistake sheds a lot of doubt to the reader, when approaching the sheer amount of calculations and formulae typical in such variational works. 2 - Linked to the above and in terms of reproducibility of the results. It is good to see in Section 3 that differential equations for the entropy and energy simplify to previously presented results in the literature. However, the bit that justifies acceptance in my opinion is the structure learning component in Section 4. In this case, pretty much all derivations are left to the appendices, the thin is rushed and feels hard to parse. The average reader will likely never look at appendices and this reduces accessibility. While it is understandable that not everything can fit, it would be valuable to add further analytic intuition behind formulae in (7)-(8). Finally, there exist minor typos that warrant a re-read. ----------- After rebuttal: I appreciate the clarifications from the authors and still think this is an above average contribution likely to spark interest. In my opinion 6 is still an appropriate score.

Reviewer 2



The paper present a variational approximation for CTBN structure learning where the variational approximation incorporates parents. I am familiar with both CTBNs and variational approximations but not the literature at the intersection; I have not checked the details of the derivations provided in the appendix. The paper develops exposition on the star variational approximation, extending mean field approximation [5] by inclusion of parents U with an approximation to the ELBO. Lines 99-101 state this explicitly. I think this could be strengthened with argumentation about (1) the quality of the approximation or (2) contrasting the approach with the mean field approach. Quality + The mathematical formulation is explicit with main results clearly described and annotated. - Some of the assumptions made are not well justified. For example, under what conditions can we assume the data likelihood of the observation model Y|X factorizes? Practically is this all/most/few/none? The same line of questioning goes for the product of gammas for the variational structure score. - The statement "only molecular biological network with a ground truth" should be amended--the field of biochemistry has tomes of known molecular networks. Clarity + the work is presented clearly Originality + the work appears original and a useful extension to previous mean field approximations. Significance + the experiments are conducted on simulated data with ground truth [amended] - no experiments using real data are conducted. **The term "synthetic" makes the analysis of the published real-world data sound like simulated data; I suggest removing it for an ML audience.** - the work is limited by assumptions and is tested on small networks (<20 nodes) despite statements of scalability. [resolved] Should the RHS terms in eq 3 be flipped? --- I have read the authors' response. I remain at a 6: addressing the underlying motivations behind the comments and questions posed would lead me to higher score.

Reviewer 3



This work proposes to improve variational approximation for continue time Bayesian networks (CTBN) using the Kikuchi functional. This approximation in turn, allows efficient Bayesian structure learning using Stirling approximation to the Gamma function. The problem of structure learning of dynamic systems is notoriously hard and has been used to study very small gene regulatory networks. The proposed solution seems to allow a significant scale up of learned network size, as it involves local computations. The Stirling approximation is an elegant solution. The authors refer to previous works that extend mean-field approximations to higher order, however more details about the relation to these works should have been given. The paper is clearly written, and required background is given where is needed to make the paper self-contained. Some clarifications are needed: (1) In the equation in line 95 the dependency of the reminder on epsilon is not clear; (2) The marginals in the equation in line 101 seem to depend on t, this should be reflected in the notation. (3) It sis not clear in line 131 why should m_n(x,0) = 0.5. The experimental section contains a comparison to a mean-field algorithm as well as to additional methods for reconstruction of regulatory network structure. To give a more comprehensive view, the authors should have compared accuracy and run-time to the method of [Studer et al. 2016] which they state shares similarities to the proposed method. The synthetic experiments contain 5 repetitions. This should be increased. --- After Author's feedback --- The authors addressed most of the comments. Relation to previous work should be written more clearly. I am keeping the original score I have given.